# OpenReview forum: "Implementing Long Text Style Transfer with LLMs through Dual-Layered Sentence and Paragraph Structure Extraction and Mapping"
_ICLR.cc/2026/Conference — ICLR 2026 Conference Withdrawn Submission_

### Official Review · Reviewer_UGRs · 2025-10-26

**Soundness:** 2
**Presentation:** 2
**Contribution:** 2
**Rating:** 2
**Confidence:** 4

**Summary:**

This paper proposes a prompting-based method for style transfer of long texts using large language models (LLMs). The main idea works in three steps: (1) first extract sentence and paragraph JSON representations from a set of reference style texts; (2) Rewrite individual sentences from input using style exemplar representations extracted from step 1; (3) Perform a refinement step to ensure paragraph-level coherence and overall style consistency.

Automatic + human evaluation (both pointwise and pairwise) show that the proposed method is slightly better than direct prompting baselines and sentence-only exemplar baselines.

**Strengths:**

1. These days users want precise control over how their LLM chatbots sound stylistically (CharacterAI, Gemini Gems etc.), which makes long-form style transfer an important and timely research problem.

2. The paper uses a variety of evaluation strategies to validate their method: (1) a joint automatic-human pointwise evaluation including human experts; (2) an automatic pairwise comparison between close systems to get finer-grained judgements. On these evaluations, the authors show consistent improvements over baselines which (1) do not use any style exemplars; (2) only use sentence-level exemplars.

**Weaknesses:**

1. **The paper is severly lacking in qualitative examples, which makes contributions very unclear**: The authors should be more precise about the kinds of style transfer tasks they are tackling, and give qualitative examples for why a paragraph-level representation is essential. Right now there is just 1 qualitative example in the Appendix, and it leaves the problem statement quite unclear to me:

```
Style Input Document: ”Bayesian optimization achieves 92% accuracy. This outperforms random search by 15%. Our method reduces computation time significantly.

Content Input Document: Neural networks reach 98% precision. They exceed logistic regression by 12%. Our approach cuts training costs substantially.

Output Document: Neural networks achieve 98% precision. This outperforms logistic regression by 12%. Consequently, our approach reduces training costs substantially while maintaining high accuracy.
```

Based on this example, I have several major concerns about the work: (1) These are far from "long-form" texts, just 3 sentence long inputs; (2) The only stylistic element the output document is using from the style input is the words "achieve" and "outperform". Why is paragraph-level structure needed for this? (3) The style and content input documents are extremely similar in style already, I doubt users will be asking questions like this. Is this representative of the actual dataset used in the work?

2. **Direct prompting with examples should suffice**: I'm quite surprised that a multi-step pipeline is necessary for this task (especially the one in Weakness 1 above), and the LLMs cannot do it directly with some few-shot examples and their excellent long-context capabilities. What was the prompt and LLM used for DirectPrompt in Figure 2? Was thinking used in DeepSeek-R1, and how do SoTA thinking models fare (like o3, Gemini 2.5 Pro, GPT5)? Note that semantic preservation has gone down for the StructuredRewritten vs DirectPrompt, which is a bit concerning.

3. **Global averaging in pointwise evals is problematic**: The pointwise automatic/human evals use an average across style/semantic/fluency scores to get the final ranking. This is quite problematic since systems could potentially hack the metric by doing well on one aspect and poorly on others on different sets of examples, see Sec3.2 in https://arxiv.org/pdf/2010.05700 for a detailed discussion on this.

4. **Very confused about methodology, are embeddings used or JSON strings?** The paper's main body mentions the use of dense sentence and paragraph embeddings (Section 4.1), but the prompts in the Appendix A are extracting JSON string templates instead (L625-L685). In the case embeddings are used, how are they fed into the pretrained LLMs? Are the vectors tokenized into multiple tokens? Is the LLM expected to calculate vector distances like in L676? If embeddings are not being used, please correct the main body and add more details on the structure of the JSON templates. Also, is it correct that no models are trained in this work, and everything is done via prompting?

5. More nit-picky, but I think these days users prefer to provide stylistic guidance using instructions rather than exemplars (although I can see both being used). One concern I have with this method is that it's too specific to the use of exemplars. A simple prompting strategy maybe more flexible and allow users to mix-and-match instructions + exemplars.

**Stylistic / Nits**

The introduction + related work are quite long (almost 4 pages!) and don't have diagrams to illustrate the method. Would be nice to shorten this and get to the contributions of the paper.

No need for equations in L237-L248 since not used again.

The paper is severly lacking in some details about the evaluation setup (L341, how is model-based eval done?), stats about the datasets used, and modeling details (see point on embedding vs non-embedding in weaknesses above).

**Questions:**

N/A, see the weaknesses above

---

### Official Review · Reviewer_Vtso · 2025-10-31

**Soundness:** 2
**Presentation:** 2
**Contribution:** 2
**Rating:** 2
**Confidence:** 4

**Summary:**

The paper targets zero-shot long-text style transfer and proposes ZeroStylus. It is a two-phase hierarchical pipeline: (1) template acquisition: LLMs extract and cluster reusable sentence patterns and paragraph structures to build repositories (2) template-guided generation: for each source paragraph, the system matches a paragraph template and then rewrites each sentence with the closest sentence template and finally refines paragraph coherence. A key design is the decoupling of sentence vs paragraph level matching and a length constrained iterative rewriting strategy to reduce style drift on long inputs. Experiments on academic text show higher style consistency and improved content preservation over direct prompting and dialogue style transfer baselines. Ablations indicate that adding paragraph templates boosts semantic preservation vs sentence-only variants.

**Strengths:**

- Clear hierarchical formulation with concrete phases, repositories, and matching criteria (incl. clustering and thresholding to control template growth).

- Practical pipeline for long texts: dual matching (sentence + paragraph), refinement, and bounded-context rewriting specifically aimed at preventing mid-document style drop-off.

- Evidence of benefit: StructuredRewritten (full pipeline) generally improves style consistency while maintaining better content preservation than sentence-only variants.

- Task is well-motivated by limitations of sentence-level methods and by the need to capture paragraph-level rhetorical flow.

**Weaknesses:**

- Evaluation dependence on LLMs: The tri-axial score uses paragraph-embedding similarity and LLM-assisted judgments. The same class of LLMs (GPT-4o, DeepSeek-R1) is also used for extraction and rewriting, raising bias and leakage concerns. Stronger human-only evaluation or cross-model evaluators would help. Moreover, existing style transfer evaluation frameworks [1] are not used.

- Baselines comparisons omit strong long-form controls (e.g., hierarchical planning/prompting, retrieval-guided author imitation, document-level rewriters with discourse constraints).

- Figure 2 shows averaged scores, but variance and significance tests are missing.

- While prompts and workflow examples are provided, exact clustering settings, repository sizes, prompt variants, paragraph windowing, and cost (LLM tokens/time) are not comprehensively listed.

- Potential content leakage: Using reference texts plus template matching invites subtle copying and safeguards (e.g. n-gram overlap audits) aren’t reported. The metric includes “keyword retention,” but plagiarism/overlap checks are not discussed.



[1] Phil Ostheimer, Mayank Nagda, Marius Kloft, and Sophie Fellenz. 2024. Text Style Transfer Evaluation Using Large Language Models. In Proceedings of the 2024 Joint International Conference on Computational Linguistics, Language Resources and Evaluation (LREC-COLING 2024), pages 15802–15822, Torino, Italia. ELRA and ICCL.

**Questions:**

- Can the authors rerun evaluation with a different LLM and a standard pipeline as in [1]?


- Can the authors provide variance/error bars and significance study for their experiments?


[1] Phil Ostheimer, Mayank Nagda, Marius Kloft, and Sophie Fellenz. 2024. Text Style Transfer Evaluation Using Large Language Models. In Proceedings of the 2024 Joint International Conference on Computational Linguistics, Language Resources and Evaluation (LREC-COLING 2024), pages 15802–15822, Torino, Italia. ELRA and ICCL.

---

### Official Review · Reviewer_npBn · 2025-11-01

**Soundness:** 3
**Presentation:** 2
**Contribution:** 2
**Rating:** 6
**Confidence:** 4

**Summary:**

This paper introduces a zero-shot framework for long-text style transfer using LLM. The proposed method, ZeroStylus, first extracts hierarchical sentence and paragraph templates from reference texts, then performs template-guided rewriting to preserve both style consistency and content integrity.

**Strengths:**

1.The idea of introducing a hierarchical template-matching mechanism for zero-shot long-text style transfer is novel and differentiates the work from existing sentence-level approaches.
2.The proposed two-stage framework effectively addresses structural coherence in long-form text, showing quantitative improvements over standard LLM style-transfer baselines.

**Weaknesses:**

1.The framework is only evaluated on academic-style writing
2.The overall pipeline is complex and depends on multiple sequential prompt calls. This limits reproducibility and makes real-world deployment difficult.
3.The paper lacks full-document coherence evaluation, which is critical when assessing LLM rewriting accuracy in long-text settings.

**Questions:**

1.The description of Phase 1 in Figure 1a and 1b is unclear. Could the authors provide a more explicit example of how sentence and paragraph templates are extracted and matched?
2.In the last part of Section 5.3 (Adversarial Evaluation), the result seems inconsistent with Table 2
3.There is no quantitative or visual analysis of the embedding clusters. Could the authors provide more concrete details or visualizations to clarify the data flow?
4.The paper introduces a style intensity parameter α, but its effect is not validated. Could you show how varying α influences the results?

---

### Official Review · Reviewer_yo8d · 2025-11-01

**Soundness:** 2
**Presentation:** 3
**Contribution:** 2
**Rating:** 4
**Confidence:** 3

**Summary:**

This paper introduces a zero-shot framework for long-text style transfer using LLMs. Unlike prior sentence-level approaches, ZeroStylus employs a dual-layered hierarchical design that integrates sentence-level stylistic adaptation with paragraph-level structural coherence. The method extracts reusable sentence and paragraph templates from reference texts, then performs template-guided rewriting to achieve style transformation while preserving content and logical flow. Experiments across academic writing datasets outperforms baseline and ablated variants in style consistency, content preservation, and expression quality, establishing approach to coherent long-form style transfer without requiring fine-tuning or parallel corpora.

**Strengths:**

This paper presents long-text style transfer using LLMs in a zero-shot setting. The proposed framework introduces a dual-layered hierarchical design that separates sentence-level stylistic adaptation from paragraph-level structural coherence, non-trivial contribution beyond prior sentence-centric TST methods. The methodology is described with formal definitions, prompt examples, and theoretical analysis. Results demonstrate improvements in style consistency, content preservation, and expression quality across multiple baselines, supported by human and model-based evaluations.

**Weaknesses:**

1. The evaluation setup relies on subjective or indirect measures of stylistic quality and could benefit from more transparent statistical analysis or error decomposition.
2.The baselines are reasonable but not exhaustive; comparisons with recent open-source document-level TST or retrieval-based rewriting systems would strengthen claims.
3. Implementation details such as template clustering thresholds, embedding models, and ablation granularity are insufficiently detailed for reproducibility.
4. The mathematical proofs, though formal, may be overly theoretical without clear empirical validation, may help more with the validation of derived bounds or with other established theretical foundations.
5. Limited evidence for applicability to other long-text domains.
6. No clear quantitative statistical tests (e.g., significance testing, confidence intervals) are reported to substantiate observed performance gains.
7. The hybrid human–LLM evaluation pipeline need more details, a more explanation would have helped more with inter-annotator reliability metrics Fleiss.
8. The style intensity parameter lacks clear calibration methodology or interpretability analysis.
9. The ablation studies are not suffiecient, evaluating only binary component removals rather than incremental or interaction effects.
10. There is no exploration of error propagation through template mismatches or failed paragraph encoding.

**Questions:**

1. How sensitive is ZeroStylus to the quality and quantity of reference style texts?
2. Could the authors clarify whether template repositories are reusable across domains, or need to be reconstructed for each style domain?
3. How are the hyperparameters (e.g., clustering ε, style intensity α) selected or tuned with more specifications?
4. Can the authors provide examples of failure cases or situations where hierarchical matching may introduce semantic drift?
5. Would incorporating retrieval-augmented LLM prompting or multi-agent refinement further enhance coherence in long documents?

---

### Note · Authors · 2025-12-08

**Comment:**

We sincerely thank all reviewers for their valuable feedback and insightful suggestions. We recognize that there remain several aspects of the paper that could be further strengthened, including the design of datasets, the enrichment of evaluations and comparison with other methods in more benchmarks, more detailed case study, as well as the overall presentation of the work. We plan to incorporate the reviewers’ suggestions to make the paper more rigorous and competitive for a future submission. We express our respect and heartfelt gratitude to all reviewers.

**Withdrawal Confirmation:**

I have read and agree with the venue's withdrawal policy on behalf of myself and my co-authors.